# The Clinical Characteristics, Microbiology and Risk Factors for Adverse Outcomes in Neonates with Gram-Negative Bacillary Meningitis

**DOI:** 10.3390/antibiotics12071131

**Published:** 2023-06-30

**Authors:** Mei-Chen Ou-Yang, Ming-Horng Tsai, Shih-Ming Chu, Chih-Chen Chen, Peng-Hong Yang, Hsuan-Rong Huang, Ching-Min Chang, Ren-Huei Fu, Jen-Fu Hsu

**Affiliations:** 1Division of Pediatric Neonatology, Department of Pediatrics, Chang Gung Memorial Hospital, Taoyuan 33382, Taiwan; oymj@cgmh.org.tw (M.-C.O.-Y.); kz6479@cgmh.org.tw (S.-M.C.); charllysc@cgmh.org.tw (C.-C.C.); qbonbon@cgmh.org.tw (H.-R.H.); rkenny@cgmh.org.tw (R.-H.F.); 2College of Medicine, Chang Gung University, Taoyuan 33302, Taiwan; mingmin.tw@yahoo.com.tw (M.-H.T.); ph6619@cgmh.org.tw (P.-H.Y.); jingmy36@cgmh.org.tw (C.-M.C.); 3Division of Neonatology and Pediatric Hematology/Oncology, Department of Pediatrics, Chang Gung Memorial Hospital, Yunlin 63812, Taiwan; 4Division of Neonatology, Department of Pediatrics, Chang Gung Memorial Hospital, Chiayi 33382, Taiwan

**Keywords:** Gram-negative bacilli, bacterial meningitis, neurological sequelae, late-onset sepsis, *E. coli*

## Abstract

**Background**: We aimed to describe the clinical features of Gram-negative bacillary (GNB) meningitis in neonates and investigate the risk factors associated with final adverse outcomes of neonatal GNB meningitis. **Methods**: From 2003 to 2020, all neonates (aged ≤ 90 days old) with bacterial meningitis who were hospitalized in four tertiary-level neonatal intensive care units (NICUs) of two medical centers in Taiwan were enrolled. Neonates with GNB meningitis were compared with those with Streptococcus agalactiae (group B streptococcus, GBS) meningitis. **Results**: During the study period, a total of 153 neonates with bacterial meningitis were identified and enrolled. GNB and GBS accounted for 40.5% (*n* = 62) and 35.3% (*n* = 54) of all neonatal bacterial meningitis, respectively. In neonates with GNB meningitis, the final mortality rate was 6.5% (4 neonates died); 48 (77.4%) had neurological complications, and 26 (44.8%) of 58 survivors had neurological sequelae at discharge. Although the final outcomes were comparable between neonates with GNB meningitis and those with GBS meningitis, neonates with GNB meningitis were more likely to have more severe clinical manifestations initially and have ventriculomegaly at follow-up. After multivariate logistic regression analysis, neonates with seizure at onset, early onset sepsis, and requirement of surgical intervention for neurological complications were independently associated with final adverse outcomes. **Conclusions**: GNB meningitis was associated with a high risk of neurological complications and sequelae, although it did not significantly increase the final mortality rate. Close monitoring of the occurrence of neurological complications and advanced therapeutic strategies to optimize the outcomes are urgently needed in the future.

## 1. Introduction

Bacterial meningitis in neonates is associated with high mortality and morbidity rates [1,2]. *Streptococcus agalactiae* (Group B *Streptococcus*, GBS) and *Escherichia coli* are the predominant pathogens that cause 65–78% of all neonatal meningitis, with high mortality rates ranging from 8–13% and 14–25% in term-born and preterm infants, respectively [2,3,4,5,6]. Neonates with bacterial meningitis are associated with a high risk of neurological complications and long-term neurological sequelae [5,6,7,8]. Recent reports have found that 35 to 55% of survivors who experience bacterial meningitis during their neonatal period will have various degrees of neurodevelopmental delay at preschool age [9,10].

Most studies regarding neonatal meningitis focus on the microbiology, epidemiology, clinical manifestations and outcomes of GBS diseases, and fewer studies look at *E. coli* meningitis [3,4,5,6,7,8,11,12,13]. Little is known about the differences between GBS meningitis and Gram-negative bacillary (GNB) meningitis in neonates, although some studies have found some uncommon GNB species as the cause of neonatal meningitis [11,12,13]. The neurological sequelae and complications after GNB meningitis are supposed to be different from those caused by GBS or other Gram-positive pathogens in neonates and young infants [1,11,12,13]. Another concern is the emergence of antibiotic-resistant GNB species, especially multidrug-resistant GNB pathogens, which will be a therapeutic challenge and lead to an increased risk of final mortality in neonates with meningitis [14,15]. Updated information on the clinical characteristics and risk factors for final adverse outcomes in neonates with GNB meningitis is important for clinicians to develop therapeutic strategies and optimize the outcomes [16,17,18]. In this study, we aimed to describe the clinical features, complications, therapeutic strategies, and outcomes of GNB meningitis in neonates and compare them with those of GBS meningitis in two tertiary level medical centers in Taiwan.

## 2. Methods

### 2.1. Patients, Study Design and Settings

Between January 2003 and December 2020, all cases of documented bacterial meningitis in neonates aged less than 90 days old were retrieved from the neonatal intensive care unit (NICU) database of Chang Gung Memorial Hospital (CGMH). The Linkou and Kaohsiung CGMHs have a total of six NICUs, and both are tertiary-level medical centers in Taiwan with a total capacity of 80 beds equipped with ventilators and 90 beds in special care nurseries. The overall annual admission in the NICUs of CGMH was approximately 1100 critically ill and preterm neonates. Only cases of primary bacterial meningitis were enrolled, and central nervous system (CNS) infections after implantation of artificial devices, including ventriculoperitoneal shunt, extraventricular drainage, and postneurosurgical CNS infections, were excluded from the analyses. The electronic records containing patient demographics, clinical manifestations, hospital courses, treatments, neurological complications, and sequelae were all reviewed and recorded. This study was approved by the Institutional Review Board of CGMH (the certificate no. 202201668B0), and a waiver of informed consent for anonymous data collection was approved.

### 2.2. Definitions and Data Collection

Meningitis was defined based on the criteria of the World Health Organization: the presence of positive cerebrospinal fluid (CSF) cultures for bacterial pathogens, plus clinical signs and symptoms of compatible central nervous system infection [19,20]. In cases with a negative CSF culture for bacterial pathogens, the following were considered to indicate bacterial meningitis: if the presence of clinical manifestations was compatible with bacterial meningitis plus a positive blood culture, and if CSF results showed at least one individual marker of bacterial meningitis, including a leukocyte count of more than 2000/μL, a glucose level of less than 34 mg/dL (1.9 mlol/L), a ratio of CSF glucose to blood glucose of less than 0.23, or a protein level of more than 220 mg/dL [21]. We applied the definitions of the Centers for Disease Control and Prevention to define severe sepsis, septic shock, and complicated bacteremia [22].

The presence of neurological complications and long-term neurological sequelae in these patients was evaluated based on the definitions described in our previous studies [23,24]. All neonates with bacterial meningitis were clinically followed by the attending physicians to assess any new neurological symptoms and signs, as well as abnormalities on neuroimaging studies, including transcranial ultrasound, computed tomography [CT] scan, or magnetic resonance imaging [MRI]. For patients with clinical signs of potential neurological complications, the date of neurological imaging was considered as the onset of events. In our institute, transcranial sonography was routinely performed for all patients with bacterial meningitis, and brain CT or MRI was conducted according to the attending physician’s decision when clinical deterioration, seizure, or abnormal neurological symptoms were present, or abnormal findings were noted in sonography. The neurological complications included the following:(1)Seizure: any symptoms of subtle seizures, including random or roving eye movements, sucking, and unusual bicycling or pedaling movements of the legs were observed. We considered seizure as a neurological complication when the observed clinical symptoms and/or abnormal epileptiform discharge were recorded on the electroencephalography (EEG) examination in neonates with bacterial meningitis without previous neurological or metabolic disorders;(2)Post-infectious encephalopathy was defined as when neonates had persistent consciousness changes for more than 24 h after the episode of meningitis;(3)Hydrocephalus and/or ventriculomegaly in neonates without previous pathology documented by transcranial ultrasound or CT/MRI after the onset of meningitis;(4)The presence of any new focal infections, including subdural empyema, arachnoiditis, ventriculitis, and spinal or brain abscesses;(5)Other neurologic complications included postmeningitis encephalomalacia, periventricular leukomalacia, brain atrophy, and cerebral infarction.

All patient demographics, symptoms, and signs at onset of meningitis, laboratory findings at onset of symptoms, treatment courses, neurological findings at discharge, and final outcomes were recorded. In our institute, neurological examinations were performed for all surviving patients by a neurologist after discharge, and the outcome was graded according to the validated Pediatric Version of the Glasgow Outcome Scale (GOS-E Peds) [25]. A favorable outcome was defined as a GOS-E Peds score of 5 (good recovery) and an unfavorable outcome was defined as a score of 1 (indicating death) to 4 (moderate disability). In the cohort, all in-hospital mortality cases and those with critical discharge on request were considered unfavorable outcomes.

### 2.3. Antimicrobial Susceptibility Testing

Antimicrobial susceptibility testing was performed with the disc diffusion method as described in previous studies [26]. For Gram-negative bacteria, antibiotic susceptibility patterns were determined according to methods recommended by the National Committee for Clinical Laboratory Standards Institute (CLSI) for the disk diffusion method and categorical assignment was carried out using CLSI breakpoints [27]. All GBS isolates were tested for susceptibility to seven antibiotics, including erythromycin, penicillin, clindamycin, vancomycin, ampicillin, cefotaxime, and teicoplanin according to the guidelines of the CLSI for the disc dilution method [27]. 

### 2.4. Statistical Analysis

We categorized neonatal bacterial meningitis into two subgroups: GNB meningitis and GBS meningitis. Neonatal meningitis caused by other Gram-positive bacteria was not analyzed in this study. Categorical and continuous variables are expressed as proportions and the median (interquartile range, IQR), respectively. Categorical variables were compared using the χ^2^ test or Fisher’s exact test; odds ratios (ORs) and 95% confidence intervals (CIs) were calculated. Continuous variables were compared using the Mann–Whitney *U*-test and *t* test, depending on the distributions. The trend of proportions of the categorical variables among the subgroups was analyzed using the Cochran–Armitage trend test. *p* values of <0.05 were considered statistically significant.

We aimed to investigate the independent risk factors for final adverse outcomes of neonates with bacterial meningitis, which included cases of final in-hospital mortality and neonates with neurological sequelae at discharge. Associations between patient demographic, clinical, and microbiological characteristics and laboratory results were tested in univariate analyses, and odds ratios (ORs) were used to quantify the strength of association. Covariates presumed to be associated with final adverse outcomes based on previous studies and those associated with mortality at *p* < 0.1 were subsequently entered into the multivariable logistic regression model. All statistical analyses were conducted using SPSS version 21.0 (IBM SPSS Statistics).

## 3. Results

### 3.1. Patient Demographics, Microbiology, and Clinical Features

During the study period, 153 neonates with bacterial meningitis were identified from the NICUs of the two medical centers in Taiwan. The mean gestational age and birth body weight in the cohort were 36.9 ± 3.54 weeks and 2754 ± 728.4 g, respectively. Most of the neonates (74.5%, *n* = 114) were full term (GA ≥ 37 weeks), only 5.9% (*n* = 9) were extremely preterm (GA ≤ 28 weeks), and 9.2% (*n* = 14) were very low birth weight (BBW < 1500 g) infants. The median age at the first positive CSF culture and diagnosis of bacterial meningitis was 30 (13.0–56.0) [interquartile range (IQR)] days old. There were 27 (17.6%) cases of early onset diseases (EOS, onset at age ≤ 7 days old) and 126 (82.4%) cases of late-onset diseases (LOS, onset at age > 7 days old and ≤ 90 days old). Most patients were inborn, and 24.8% (*n* = 38) were transferred from other hospitals. The patient demographics were comparable between neonates with GNB meningitis and those with GBS meningitis and are summarized in Table 1.

In total, Gram-negative bacteria accounted for 40.5% (*n* = 62) of neonatal bacterial meningitis and 35.3% (*n* = 54) of neonates with acute bacterial meningitis were caused by group B *Streptococcus* (GBS). The predominant Gram-negative bacteria of neonatal meningitis included *E coli* (22.2%, *n* = 34), *Klebsiella* spp. (6.5%, *n* = 10), *Serratia marcescens* (3.9%, *n* = 6), *Salmonella* (2.6%, *n* = 4), and *Pseudomonas* spp. (2.6%, *n* = 4). Other GNB pathogens were *Enterobacter* spp. (1.3%, *n* = 2), *Morganella morganii* (*n* = 1), and *Citrobacter koseri* (*n* = 1). A total of 21 (13.7%) cases of neonatal meningitis were caused by Gram-positive cocci other than GBS, and the most common was *Staphylococcus aureus*. A total of 11 patients had negative CSF cultures but at least one individual CSF marker and clinical symptoms of bacterial meningitis. Five patients had CSF cultures positive for *Candida* spp.

In our cohort, 128 (83.7%) neonates had concurrent bacteremia, and all of their blood cultures grew the same pathogen as that in their CSF culture. The majority of the symptoms were typical of neonatal sepsis, and the initial presentations were comparable between neonates with GNB meningitis and those with GBS meningitis. However, neonates with GNB meningitis were more likely to have apnea, cyanosis, and abdominal distension than those with GBS meningitis (*p* = 0.045 and 0.028, respectively) (Table 1). Additionally, neonates with GNB meningitis were significantly more likely to require oxygenation support, present with severe sepsis, and have thrombocytopenia (*p* = 0.031, 0.015, and 0.029, respectively). Lumbar puncture was performed on the same day when neurological symptoms were noted or the blood culture showed positive growth of GNB and/or GBS in all patients, although most of the GNB meningitis cases were hospital acquired when compared with cases of GBS meningitis (Table 1). Of note, a relatively higher glucose level, a significantly lower protein value, and lower white blood cell counts were noted in neonates with GNB meningitis than in those with GBS meningitis (Table 1).

All cases were treated with ampicillin plus third-generation cephalosporin, mostly cefotaxime, as the initial empiric antibiotics, while some of them had ampicillin plus cabapenem as the initial treatment, depending on the decision of the attending physicians. Among neonates with GNB meningitis, 34 patients experienced clinical deterioration within 12 h after initiation of empiric antibiotics, although transtentorial cerebral herniation with pupil dilation or abnormal posturing was never noted in the cohort. The clinical deterioration consisted of septic shock in 25 patients, respiratory failure in 34, new onset of seizure in 12, and a decreased level of consciousness in 7. All patients had a cranial sonography examination following bacterial meningitis, and cranial imaging was performed in 42 (67.7%) of 62 patients, mostly due to seizure and lethargy or irritability. The neurological complications of neonates with GNB meningitis are summarized in Table 2.

### 3.2. Therapeutic Strategies and Final Outcomes

Among Gram-negative bacteria, only 7 were multidrug-resistant pathogens. Therefore, most patients (*n* = 56, 90.3%) received microbiologically adequate antibiotics within 24 h after onset of meningitis. Surgical intervention was performed in 16 patients, including extraventricular drainage (*n* = 9), subdural-peritoneal shunt (3), and ventriculoperitoneal shunt (*n* = 7), and 6 had bilateral subdural drain. The therapeutic duration of neonates with GNB meningitis was significantly longer than that of neonates with GBS meningitis (the median duration of antimicrobial treatment was 26 days in surviving patients versus 17 days in neonates with GBS meningitis, *p* < 0.05). Four patients (6.5%) died and three were discharged at their family’s request at critically ill conditions. However, 77.4% (*n* = 48) had neurological complications during hospitalization (Table 2), and 44.8% (26/58) of the survivors had neurological sequelae at discharge.

We also investigated the independent risk factors for final adverse outcomes, defined in this study as those with final mortality and neurological sequelae at discharge (Table 3). Preterm neonates (GA < 37 weeks) had a significantly higher risk of worse outcomes than term-born neonates. Neonates with early onset disease, seizure at onset (defined as seizure attack within 48 h after onset of meningitis), respiratory failure, and septic shock, and neonates with neurological complications who required surgical intervention, were enrolled into the multivariable regression model for assessment of the significant association with final adverse outcomes. After multivariate logistic regression analyses, the independent risk factors for final adverse outcomes in neonates with acute bacterial meningitis were seizure at onset (OR, 2.40; 95% CI: 1.12–4.14, *p* = 0.025), early onset sepsis (OR, 3.35; 95% CI: 1.09–8.29, *p* = 0.035), and requirement of surgical intervention for neurological complications (OR, 3.34; 95% CI: 1.78–6.86, *p* < 0.001).

## 4. Discussion

To the best of our knowledge, this is the first study to compare the differences between neonates with GNB meningitis and those with GBS meningitis [11], and we found that the final outcomes were comparable between GNB and GBS meningitis. *E. coli* was the predominant pathogen, but other GNB microorganisms were not uncommon. In our cohort, 77.4% of neonates with bacterial meningitis had neurological complications and 44.8% of the survivors had neurological sequelae at discharge, although the mortality rate for neonatal meningitis was not significantly higher than the reported 8.9–15% mortality rate of neonates with late-onset sepsis [28,29,30]. Some of the neurological complications occurred several weeks after the onset of neonatal meningitis, which highlights the importance of continuous monitoring and high alertness for relevant symptoms and signs. After multivariate logistic regression analysis, EOS, requirement of surgical intervention for neurological complications, and seizure at onset of meningitis were independently associated with final adverse outcomes.

The pathogen distributions of neonatal meningitis in our cohort were consistent with those reported in previous studies [4,5,6,8,12], in which *E coli* and GBS were the predominant causes. However, some studies found *Klebsiella pneumonia*, *Acinetobacter baumannii*, *Serratia marcescens*, and *Enterobactor* spp. to be important pathogens of neonatal GNB meningitis [31,32,33], especially in low- and lower–middle-income countries [32,34]. We suspected the sources of most GNB meningitis to be from the gastrointestinal tracts, where the pathological bacteria will penetrate the gut mucosa and cause bacteremia in immunocompromised hosts, such as long-term hospitalized neonates [31,33]. In Taiwan, current infection control and prevention strategies have worked successfully to reduce vertical transmission from mothers and contaminated hospital environments [35]. We found that the incidence of GBS EOS has declined in the past decade, and it is known that EOS, especially in neonates with meningitis, is highly associated with worse outcomes [28,29,30]. Additionally, the antibiotic resistance rate of Gram-negative bacteria in our cohort was significantly lower than that in other countries; therefore, initial inadequate antibiotic therapy was uncommon and not related to worse outcomes [32,34]. These factors may account for the relatively lower mortality rate in our cohort. Therefore, early identification and prevention of neurological sequelae in neonates with bacterial meningitis should be the focus in the future.

A high and comparable rate of neurological complications was noted in neonates with GBS and GNB meningitis, which can explain the longer treatment duration in our cohort. Additionally, the therapeutic strategies may also be response related. Direct bacterial invasion and sepsis-associated encephalopathy are two potential mechanisms that cause neurological complications in the acute and subacute stages, respectively [23,36,37]. We have previously concluded that cerebral hypoperfusion due to septic shock and GBS infection were associated with a 5.9- and a 6.8- to 8.9-fold increased risk of developing neurological complications in neonates with complicated sepsis and meningitis, respectively [23,24]. Other studies have also found that seizures at the onset of meningitis, higher illness severity, and some specific pathogens were independently associated with neurological sequelae and final adverse outcomes [3,23,24]. However, we could not find significantly higher mortality or morbidity rates of specific GNB pathogens or GBS isolates, which may be due to inadequate case numbers. 

We found that the requirement of surgical intervention was associated with final adverse outcomes. This can be explained by the presence of neurological complications and an increased risk of another episode of nosocomial infection. This result also highlights the importance of avoiding the occurrence of neurological complications. In our cohort, although neonates with GNB meningitis had significantly lower protein levels and higher glucose values in the CSF examinations than those with GBS meningitis, this cannot be linked to the severity of intracranial inflammation because the reference values vary greatly in the neonatal period [38,39]. Of note, some neurological complications, including ventriculomegaly, hydrocephalus, brain infarction, and encephalomalacia were often detected several weeks after the first positive CSF culture. Because hydrocephalus, brain infarction, and encephalomalacia are associated with an increased risk of long-term neurological sequelae and neurodevelopmental delay [40,41], alertness and continuous monitoring of neonates with bacterial meningitis are indicated for early intervention of neurological complications.

There are some limitations in this study. This was not a multicenter or population-based surveillance study, and only patients from two medical centers were enrolled, which limits the generalization and application of the conclusion to other institutes and countries. Because cranial images were not routinely performed and there were no standardized therapeutic strategies in our institute, some neurological complications may be missed, and there may be some biases in the study. Although the case number in this study was adequate when compared with that of some nationwide cohort studies or meta-analyses [2,3,5,12], subgroup analysis was not performed in this study. Therefore, the clinical importance of specific GNB species meningitis, such as *Salmonella* spp. or *Klebsiella* meningitis, was not investigated in this study. Additionally, there might be some CSF culture-negative patients or cases without available data that were missed during the long study period, which may lead to some bias in this study.

## 5. Conclusions

In conclusion, although bacterial meningitis does not significantly increase the mortality rate in neonates with GBS sepsis or late-onset sepsis in the NICU, neonates with acute bacterial meningitis are at a high risk of neurological complications and neurological sequelae, which cause a significant burden to the community and the family. Therefore, continuous monitoring of neonates with bacterial meningitis, especially those with GNB sepsis and complicated GBS sepsis is important in clinical practice. GNB species other than *E coli* are not uncommon as the causative pathogen of acute bacterial meningitis in neonates. Early identification of clinical signs associated with a higher risk of death and development of a predictive scoring model are warranted in the future.

## Figures and Tables

**Table 1 antibiotics-12-01131-t001:** Patient demographics and clinical features of neonates with Gram-negative bacillary (GNB) meningitis and those with group B *Streptococcus* (GBS) meningitis in Chang Gung Memorial Hospital (CGMH), 2003–2020.

	All Cases (Total *n* = 153)	GNB Meningitis (Total *n* = 62)	GBS Meningitis (Total *n* = 54)	*p* Values
Gestational age, (week)	38.0 (36.0–39.0)	38.0 (36.8–40.0)	38.0 (37.0–39.0)	0.939
Birth body weight, (g)	2890.0 (2490–3227.5)	2900.0 (2400–3220)	2940.0 (2637.5–3253.8)	0.294
Gender, (male/female, n/%)	71 (46.4)/82 (53.6)	32 (51.6)/30 (48.1)	22 (40.7)/32 (59.3)	0.267
Birth by NSD/Cesarean section, n (%)	104 (68.0)/49 (32.0)	40 (64.5)/22 (35.5)	36 (66.7)/18 (33.3)	0.814
5 min Apgar score <7, n (%)	15 (9.8)	7 (11.3)	2 (3.7)	0.128
Premature rupture of membrane, n (%)	27 (17.6)	14 (22.6)	9 (16.7)	0.244
Concurrent bacteremia, n (%)	128 (83.7)	58 (93.5)	51 (94.4)	0.904
Onset of meningitis at days of life (day), median (IQR)	26.5 (9.5–69.5)	30.0 (10.0–74.0)	22.0 (7.5–46.0)	0.128
Early onset sepsis (≤7 days), n (%)	27 (17.6)	9 (14.5)	15 (27.8)	
Late-onset sepsis (8–90 days), n (%)	126 (82.4)	53 (85.5)	39 (72.2)	
Onset of meningitis at days of hospitalization (day) median (IQR)	17.0 (2.5–22.0)	1.0 (1.0–8.0)	19.0 (4.0–38.0)	<0.001
Clinical features *, n (%)				
Fever (≥38.3 °C)	110 (71.9)	49 (79.0)	46 (85.2)	0.472
Apnea, bradycardia and/or cyanosis	91 (59.5)	48 (77.1)	32 (59.3)	0.045
Ventilator requirement				0.031
Room air	62 (40.5)	14 (22.9)	22 (40.7)	
Nasal canula	9 (5.9)	5 (8.1)	4 (7.4)	
Non-invasive ventilator (N-CPAP and N-IMV)	16 (10.4)	9 (14.5)	4 (7.4)	
Intubation	56 (36.6)	28 (45.2)	22 (40.7)	
High-frequency oscillatory ventilator	10 (6.5)	6 (9.7)	2 (3.7)	
Abdominal distension and/or vomiting	92 (60.1)	48 (77.4)	31 (57.4)	0.028
Hypoglycemia	26 (17.0)	13 (21.0)	7 (13.0)	0.327
Hypotension	54 (35.1)	25 (40.3)	14 (25.9)	0.118
Severe sepsis	73 (47.7)	42 (56.5)	24 (44.4)	0.015
Disseminated intravascular coagulopathy	31 (20.3)	20 (32.3)	10 (18.5)	0.136
Requirement of blood transfusion **	89 (58.2)	42 (67.7)	30 (55.5)	0.186
Laboratory data at onset of GBS bacteremia, n (%)				
Leukocytosis (WBC > 20,000/L)	79 (51.6)	35 (56.5)	28 (63.0)	0.578
Leukopenia (WBC < 4000/L)	52 (34.0)	26 (41.9)	18 (33.3)	0.341
Shift to left in WBC (immature > 20%)	35 (22.9)	16 (25.8)	11 (20.4)	0.514
Anemia (hemoglobin level < 11.5 g/dL)	80 (52.3)	35 (56.5)	30 (55.5)	0.853
Thrombocytopenia (platelet < 150,000/μL)	42 (27.5)	26 (41.9)	12 (22.2)	0.029
Metabolic acidosis	54 (35.3)	28 (45.2)	17 (31.5)	0.129
Coagulopathy	52 (34.0)	27 (43.5)	15 (27.7)	0.085
C-reactive protein (mg/dL), median (IQR)	121.0 (50.6–187.4)	121.2 (67.8–164.8)	103.2 (44.5–210.8)	0.516
Cerebrospinal fluid examinations				
WBC count (/L), median (IQR)	32.2 (11.5–480.0)	23.9 (8.6–372.5)	400.0 (26.5–2045.0)	0.016
Protein level (mg/dL), median (IQR)	274.6 (111.6–417.2)	231.8 (85.5–358.8)	332.0 (233.8–480.5)	0.002
Glucose level (mg/dL), median (IQR)	31.0 (7.0–52.0)	39.0 (5.0–52.5)	21.5 (7.5–35.5)	0.085

All *p* values are the comparisons between neonates with Gram-negative bacillary meningitis and those with group B *Streptococcus* meningitis. All data are expressed as numbers (%) or medians (IQRs). * At onset of bacterial bacteremia. ** Including leukocyte-poor red blood cells, platelet transfusion and correlation of coagulopathy and/or disseminated intravascular coagulopathy. IQR: interquartile range; WBC: white blood cell count; N-CPAP: nasal continuous positive airway pressure; N-IMV: non-invasive mechanical ventilation.

**Table 2 antibiotics-12-01131-t002:** Neurological complications in neonates with Gram-negative bacillary (GNB) meningitis versus group B streptococcal (GBS) meningitis in CGMH, 2003–2020.

Neurological Complications, Sequelae and Death	GNB Meningitis (*n* = 62)	GBS Meningitis (*n* = 54)	*p* Values
Any neurological complications	48 (77.4)	44 (81.5)	0.651
Seizure	19 (30.6)	33 (61.1)	0.001
Subdural effusion	29 (46.8)	28 (51.9)	0.710
Increased intracranial pressure	19 (30.6)	11 (20.4)	0.288
Ventriculomegaly	31 (50.0)	17 (31.5)	0.037
Hydrocephalus	11 (17.7)	12 (22.2)	0.644
Encephalomalacia	3 (4.8)	7 (13.0)	0.185
Subependymal hemorrhage	7 (11.3)	6 (11.1)	1.000
Intraventricular hemorrhage	6 (9.7)	4 (7.4)	0.664
Ventriculitis	6 (9.7)	7 (13.0)	0.606
Periventricular leukomalacia	4 (6.5)	6 (11.1)	0.372
Infarction	2 (3.2)	8 (14.8)	0.017
Subdural empyema or abscess	13 (21.0)	14 (25.9)	0.660
Brain atrophy	1 (1.6)	2 (3.7)	0.462
Discharge with neurological sequelae	26/58 (44.8)	19/50 (38.0)	0.588
Final in-hospital mortality	4 (6.5)	4 (7.4)	0.779

All data are expressed as number (%).

**Table 3 antibiotics-12-01131-t003:** Risk factors for final unfavorable outcomes (death or major neurological sequelae at discharge) in neonates with Gram-negative bacillary meningitis by univariate and multivariate analyses.

Parameters	Univariate Analysis		Multivariate Analysis	
	OR (95% CI)	*p* Value	Adjusted OR (95% CI)	*p* Value
Preterm birth (GA < 37 weeks)	2.28 (1.08–4.81)	0.030	1.80 (0.77–4.22)	0.178
Septic shock	2.11 (1.01–4.39)	0.047	1.94 (0.56–6.71)	0.298
Respiratory failure (requirement of intubation)	2.53 (1.12–5.74)	0.026	1.23 (0.32–4.71)	0.766
Concurrent bacteremia	0.78 (0.41–1.50)	0.462		
Gram-negative bacilli versus GBS	1.36 (0.60–3.05)	0.462		
High protein level in CSF	0.93 (0.46–1.87)	0.835		
Early onset sepsis	4.26 (1.51–12.04)	0.006	3.35 (1.09–8.29)	0.035
Requirement of surgical intervention	3.72 (1.83–7.55)	< 0.001	3.34 (1.78–6.86)	< 0.001
Seizure at onset	2.48 (1.28–4.81)	0.007	2.40 (1.12–4.14)	0.025
Thrombocytopenia (platelet count < 150,000/μL)	0.89 (0.41–1.91)	0.762		

GA: gestational age; CSF: cerebrospinal fluid; OR: odds ratio; 95% CI: 95% confidence interval.

## Data Availability

The datasets used/or analyzed during the current study are available from the corresponding author on reasonable request.

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
