# Peer review of "The Clinical Characteristics, Microbiology and Risk Factors for Adverse Outcomes in Neonates with Gram-Negative Bacillary Meningitis"

_antibiotics, 2023, doi:10.3390/antibiotics12071131_

Round 1

Reviewer 1 Report

The paper is useful given the severity of complications of Gram-negative bacillary meningitis and the paucity of individual clinician experience of the condition. The abstract describes the findings well. The review covered a long period and was retrospective, so it is likely there were changes in data recording and management affecting the results. The neurological assessment would depend on practice at the time. A positive blood culture was required for inclusion whereas it is possible that the organism would be isolated from CSF and not blood. Hence cases may be under-reported. The length of follow up for complications was not stated, although some cases of meningitis up to 90 days. CLSI breakpoints changed over time so the reporting of sensitivities would depend on the protocol at the time. The sensitivity results need to be brought to the same standard. The number of patients excluded due to missing data has not been reported.

In table 1, ‘Onset of GBS bacteremia’ has been applied even to the GNB meningitis group. Is this a mistake? A long list of factors should not use p 0.05 as a criterion of significance, it should be 0.01. What proportion of patients with septic shock were seen by a senior clinician within 1-2 h of presentation? The risk of seizure was higher in Gram negative meningitis presumably indicating the severity of the infection, but could it be inappropriate empirical treatment? The results state 90% had adequate antibiotics suggesting the empirical agents were active but what agents were used? Presumably the choice of antibiotic and use of agents to limit cerebral oedema have changed. Different regimens are likely to have affected mortality rates differently. Treatment was given for longer than other cases of meningitis but was that response related?

The information is valuable but providing these additional points would help clarify the reliability of the conclusions.

Author Response

Dear reviewer:

      Please see the attachment. I appreciate your comments and review, thank you.

Best regard,

Tsai Ming Horng

Reviewer 2 Report

The authors describe the clinical characteristics, microbiology and risk factors for adverse outcomes in neonates with meningitis caused by Gram negative bacilli. The paper is interesting however two main issues should have to be addressed:

1)         did the authors also use syndromic panel or any other molecular assay test in order to diagnose meningitis? Because the rapid diagnostic tests are worldwide extensively used and their introduction has changed the paradigm in the diagnosis of meningitis.. Culture approach cannot be sufficient anymore.

2)         The population studied are well described, but in order to justify the high prevalence of K.pneumoniae, A.baumanii, Serratia and Enterobacter (which represent typical hospital acquired pathogens). Was meningitis the cause of hospitalization or did it occur as a complication?

If meningitis occurred after the admission, it has to be evaluated another parameter: delta time between the admission to the hospital and the collection of CSF. Is it possible that those meningitis were hospital acquired infections due to presence of any devices? Were patients screened at admission for example for rectal colonization by A.baumanni or K.pneumoniae?

Author Response

(The authors gave the same response as above.)

Reviewer 3 Report

If available maternal management for Group B strep disease/colonization would be a desireable piece of information for both GBS neonates and GN neonates.

Author Response

(The authors gave the same response as above.)

Round 2

Reviewer 1 Report

Thank you for responding to comments.

Reviewer 2 Report

none